# SGLT2 Inhibitors in Diabetic and Non-Diabetic Chronic Kidney Disease

**DOI:** 10.3390/biomedicines11020279

**Published:** 2023-01-19

**Authors:** Manuel Alfredo Podestà, Gianmarco Sabiu, Andrea Galassi, Paola Ciceri, Mario Cozzolino

**Affiliations:** 1Renal Division, ASST Santi Paolo e Carlo, Department of Health Sciences, Università degli Studi di Milano, 20154 Milano, Italy; 2Transplantation Research Center, Renal Division, Harvard Medical School, Brigham and Women’s Hospital, Boston, MA 02115, USA; 3Nephrology and Dialysis Unit, ASST Fatebenefratelli Sacco, Università degli Studi di Milano, 20154 Milano, Italy

**Keywords:** empagliflozin, dapagliflozin, canagliflozin, sotagliflozin, ertugliflozin, proteinuria, renal failure

## Abstract

Results from recent randomized controlled trials on inhibitors of the sodium-glucose cotransporter 2 (SGLT2) have determined a paradigm shift in the treatment of patients with type 2 diabetes mellitus. These agents have been shown not only to ameliorate metabolic control, but also to independently protect from cardiovascular events and to reduce the progression of chronic kidney disease (CKD) in these patients. The magnitude of the nephroprotective effect observed in these studies is likely to make SGLT2 inhibitors the most impactful drug class for the treatment of diabetic patients with CKD since the discovery of renin–angiotensin system inhibitors. Even more surprisingly, SGLT2 inhibitors have also been shown to slow CKD progression in non-diabetic individuals with varying degrees of proteinuria, suggesting that activation of SGLT2 is involved in the pathogenesis of CKD independent of its etiology. As indications continue to expand, it is still unclear whether the observed benefits of SGLT2 inhibitors may extend to CKD patients at lower risk of progression and if their association with other agents may confer additional protection.

## 1. Introduction

The active process of glucose reabsorption in renal tubular cells is regulated by the activity of molecules called sodium-glucose cotransporters (SGLT). The SGLT cotransporters ensure that the approximately 200–300 g of glucose normally filtered every day at the glomerular level under physiological conditions is almost completely reabsorbed. The SGLT2 isoform, present in the S_1_ tract of the proximal tubule, is responsible for the reabsorption of more than 90% of the filtered glucose load, while SGLT1, the distal isoform found in S_2_ and S_3_ segments, has an increased affinity for glucose but with lower absorption capacity [1,2].

Selective inhibition of SGLT channels by phlorizin derivatives, called SGLT2 inhibitors (SGLT2is) or gliflozins, has profound metabolic and hemodynamic effects. Following the amazing results shown in randomized controlled trials in terms of protection from cardiovascular events [3], several drugs belonging to this class have been approved for the treatment of patients with diabetes mellitus. More recently, SGLT2is have also demonstrated marked efficacy in the prevention of chronic kidney disease (CKD) progression independent of diabetes status [4], with an effect size of such proportions as to determine a paradigm shift in the treatment of patients with kidney disease.

In this review, we summarize the available data about the systemic and renal effect of these drugs and discuss the results of the most recent studies dedicated to patients with CKD.

## 2. Nephroprotective effects of SGLT2is

In healthy subjects, the tubular glucose reabsorption capacity is almost complete up to blood glucose values of approximately 160–180 mg/dL [5]. In diabetic patients, this threshold increases to about 250 mg/dL due to the development of compensatory mechanisms consequent to the high glycemic load, which include proximal tubular hypertrophy with an increase in SGLT2 expression [6]. In patients with CKD, even in the absence of diabetes, a phenomenon of single-nephron hyperfiltration occurs, linked to the adaptive responses due to the progressive decrease in the number of functional units [7]. Consequently, even with normal glycemia, this state determines the filtration and reabsorption of an increased glycemic load for each single nephron [8]. Under these conditions, inhibition of SGLT channels results in effects that extend far beyond simple glycosuria.

Data from early clinical trials of SGLT2is have shown that these drugs reduce glycated hemoglobin levels by 0.5–1.0% without causing hypoglycemic events [9]. This occurs because the glucose reabsorption threshold at the tubular level is reduced by these drugs to values not lower than 75 mg/dL, well above the threshold for symptomatic hypoglycemia [10]. Lower glycaemia and improvement of insulin resistance lead to a reduction in microvascular complications in the long term, and the increased production of ketone bodies due to a higher glucagon/insulin ratio may improve energy utilization and mitochondrial function [11]. Additional metabolic changes contribute to the renal protective effects (Figure 1). The reduction in glucose reabsorption at the tubular level leads to a reduction in energy expenditure by the basolateral Na^+^-K^+^-ATPase to maintain the sodium gradient necessary for luminal cotransport [12]. This results in lower metabolic consumption and workload of the tubular cells, making them less susceptible to damage. Furthermore, glucose uptake by the tubular cells increases their glycolytic metabolism, which makes them more prone to epithelial–mesenchymal transition, a state that preludes fibrosis [13]. In vitro studies have also shown that high glucose levels exert direct toxicity in proximal renal tubular cells by inducing oxidative stress and advanced glycation end-products, p21-dependent senescence and production of proinflammatory and profibrotic mediators, effects that are counteracted by SGLT2 inhibition [14,15]. The inhibition of these mechanisms may therefore partly explain the protective effect of SGLT2is on the progression of renal failure, especially in diabetic patients.

However, the protective effects of SGLT2is on the kidney appear to be independent of their effects on blood sugar levels [16], and have been predominantly ascribed to modifications in renal hemodynamics. The alteration of the tubuloglomerular feedback mechanism with consequent hyperfiltration is a known driver of CKD progression in diabetic patients [17]. Decreased sodium delivery to the macula densa due to proximal tubular hypertrophy causes decreased adenosine release, resulting in vasodilatation of the afferent arteriole and increase in intraglomerular pressure [18]. Although comparatively less pathogenetically defined, CKD patients without diabetes may show several similarities with these mechanisms. As already highlighted, single-nephron hyperfiltration and hyper-reabsorption are common in CKD due to the mismatch between the number of residual functional units and metabolic demand. Moreover, alterations of tubuloglomerular feedback and glomerular perfusion autoregulation have also been described in CKD [19,20]. Thus, SGLT2 inhibition increases distal sodium delivery, which in turn promotes the tubuloglomerular feedback and leads to restoration of intraglomerular pressure. Furthermore, SGLT2i administration appears to result in adenosine-mediated efferent arteriolar dilatation, even in patients with effective inhibition of the renin–angiotensin system (RAS) [21]. These changes result in the initial GFR dip observed in patients initiating SGLT2is, which closely resembles that seen at the initiation of therapy with RAS antagonists and should be regarded as a sign of hemodynamic efficacy rather than representing a matter of concern [22]. As described below, data from trials as well as from real-world experiences [23] convincingly showed that the incidence of acute kidney injury is reduced in patients who are prescribed SGLT2is. The traditional threshold to define an acceptable creatinine rise after RAS inhibitor initiation (up to a 30% increase from baseline) seems to be a reasonable choice to inform the decision on whether SGLT2i treatment should be withheld and volume status reassessed [24].

Other pleiotropic effects of SGLT2is include an increase in natriuresis with a reduction in circulating volume [25] and a modest reduction in systemic blood pressure, typically about 4 mmHg for systolic and 2 mmHg for diastolic BP [26]. This effect on blood pressure is maintained even at advanced CKD levels, probably due to the improvement of endothelial function and the reduction in sympathetic nerve activity that are recorded early in the course of therapy [27]. SGLT2is also reduce the levels of some proinflammatory mediators, such as IL-6, TNF and C-reactive protein [28,29], and not only reduce body mass over time [26] but also modify the characteristics of perirenal fat by reducing its proinflammatory and fibrogenic capacity [30].

Overall, it is difficult to pinpoint a single mechanism responsible for the protective effect of SGLT2is on CKD progression. The most probable hypothesis is that the combination of these factors could have led to the results observed in clinical trials.

## 3. SGLT2is in Diabetic Patients

### 3.1. RCT with Primary Cardiovascular Endpoints

As a direct consequence of the 2008 FDA recommendations regarding the need to demonstrate cardiovascular safety for any new antidiabetic agent, several SGLT2is have been studied in randomized controlled trials with cardiovascular events as the primary outcome. The secondary analyses of these trials were the first to show a protective effect of SGLT2is on CKD progression.

The EMPA-REG OUTCOME trial enrolled 7020 patients with type 2 diabetes mellitus and high cardiovascular risk, with a GFR > 30 mL/min [31]. Randomization to empagliflozin not only reduced the risk of reaching the combined primary endpoint of major cardiovascular events by 14%, but exploratory analyses also showed a significant reduction (HR [95%CI]: 0.54 [0.40–0.75]) in the composite renal endpoint of doubling of serum creatinine, initiation of renal-replacement therapy or death from renal disease [32]. Consistently, treatment with empagliflozin was associated with an adjusted mean difference in eGFR compared to placebo of 4.7 mL/min/1.73 m^2^ over the 3.1-year median follow-up of the trial. These effects were independent of the drug dose, CKD stage and the severity of proteinuria.

The CANVAS program combined data from two trials enrolling a total of 10.142 diabetic patients at high cardiovascular risk, 72% of whom had a previous history of atherosclerotic vascular disease [33]. This trial also excluded patients with eGFR < 30 mL/min/1.73 m^2^, but 30% of enrolled subjects had micro- or macroalbuminuria at baseline. Allocation to canagliflozin reduced the primary composite cardiovascular endpoint by 14%, but the risk reduction in hard renal outcomes was even more striking (HR [95%CI]: 0.60 [0.47–0.77]). Consistent with the pharmacological action of canagliflozin, regression of albuminuria was observed more frequently in patients randomized to the drug compared to placebo.

The effect of dapagliflozin on cardiovascular outcomes was evaluated in the DECLARE–TIMI 58 study, the largest trial for SGLTis so far, with a total of 17.160 enrolled diabetic patients [34]. Dapagliflozin significantly reduced the incidence of cardiovascular death and hospitalizations for heart failure but did not reduce the incidence of major cardiovascular events. The trial excluded subjects with eGFR < 60 mL/min, but still reported a significant reduction in the renal composite outcome (HR [95%CI]: 0.76 [0.67–0.87]).

Though extremely informative, these trials were not designed to primarily assess the effects of SGLT2is on kidney outcomes. For this reason, the SCORED trial, which specifically enrolled diabetic patients with CKD (eGFR between 25 and 60 mL/min/1.73 m^2^), reported a reduction in the incidence of cardiovascular death, hospitalizations and urgent visits for heart failure, but failed to observe a significant effect on kidney-related outcomes [35]. Similarly, the VERTIS CV trial, although successful on the primary cardiovascular outcome, failed to observe a significant effect of ertugliflozin on renal endpoints [36]. Even in this cohort of patients, however, exploratory analyses showed that ertugliflozin had a favorable placebo-adjusted eGFR slope of more than 0.75 mL/min/1.73 m^2^ per year [37].

Several additional randomized controlled trials were conducted to evaluate the effect of SGLT2is on cardiovascular outcomes in patients with heart failure with or without diabetes (DAPA-HF [38,39], EMPEROR-Reduced [40], EMPEROR-Preserved [41], DELIVER [42] and SOLOIST-WHF [43]). These studies all showed a significant benefit of SGLT2is on primary composite cardiovascular endpoints. Moreover, though not powered for renal outcomes, secondary analyses of these trials also consistently showed a reduction in the rate of GFR decline in patients allocated to SGLT2is. Of note, this effect seemed to be present in both diabetic and non-diabetic patients in the DAPA-HF trial [39].

### 3.2. RCT with Primary Renal Endpoints

The CREDENCE study was the first trial to assess the effect of SGLT2is on kidney-related outcomes. This trial enrolled 4401 patients with type 2 diabetes and albuminuric CKD (eGFR of 30–90 mL/min/1.73 m^2^ and UACR of 300–5000 mg/g) on maximally tolerated RAS inhibitor background therapy [44]. The trial was stopped early after a planned interim analysis due to achievement of prespecified efficacy criteria: after a median follow-up of 2.6 years, canagliflozin significantly reduced the incidence of the composite outcome of end-stage kidney disease, doubling of serum creatinine or death from renal causes (HR [95%CI] 0.66 [0.53–0.81]). The effects were consistent across prespecified subgroups, which included strata based on estimated GFR and UACR levels. The change in the estimated GFR slope favored the canagliflozin arm, with a between-group difference of 1.52 mL/min/1.73 m^2^ per year. Following the publication of these results, the KDIGO guidelines for diabetes management in CKD recommended the use of SGLT2is as first-line treatment with metformin [45].

## 4. SGLT2is in Non-Diabetic Patients with CKD

Albeit groundbreaking, the CREDENCE trial could not answer the key question of whether the nephroprotection conferred by SLGT2i could extend also to CKD patients without diabetes. The DAPA-CKD trial was designed to test this hypothesis, by including 4304 diabetic and non-diabetic patients with CKD (eGFR of 25–75 mL/min/1.73 m^2^ and UACR of 200 to 5000 mg/g) receiving a stable dose of RAS blockers for at least 4 weeks before enrollment [46]. This trial was also stopped early at a median follow-up of 2.4 years, with dapagliflozin reducing the risk of reaching the primary combined endpoint (GFR decline > 50%, end-stage kidney disease, or death from renal or cardiovascular causes) by a staggering 39%, which further increased when the renal components of the outcome were considered separately (HR [95%CI] 0.56 [0.45–0.68]). These effects were consistent across levels of renal function and albuminuria, but the absolute novelty was that dapagliflozin was equally effective in non-diabetic patients, with a large effect size on the primary endpoint (HR [95%CI] 0.50 [0.35–0.72]). With 19 patients as the number needed to treat to prevent one primary outcome event, SGLT2is demonstrated to be the most effective class of drugs for the prevention of CKD progression since the discovery of RAS inhibitors. As previously observed, dapagliflozin induced an acute dip in the GFR, which thereafter stabilized, leading to a between-group difference of 0.93 mL/min/1.73 m^2^ per year in favor of the drug compared to placebo over the whole study duration. In this population, quality-adjusted life year gain due to SGLT2i use was estimated between 0.82 and 1.00 by cost-effectiveness analyses [47]. Trial-level estimates from DAPA-CKD were also used to calculate the absolute estimated benefit of combined treatment with RAS and SGLT2 inhibitors in non-diabetic patients with CKD and albuminuria [48]. For a 50-year-old patient until the age of 75 years, the drug combination provided a 7.4-year gain in kidney-failure-free survival compared to a theoretical control not receiving any treatment.

Recently published results from the EMPA-KIDNEY trial confirmed and expanded these findings [49]. This study enrolled 6609 CKD patients on background RAS inhibition with a broader range of severity, including subjects with an eGFR between 20–45 mL/min/1.73 m^2^ or those with eGFR between 45–90 mL/min/1.73 m^2^ and a UACR of at least 200 mg/g. Importantly, the trial included 3569 subjects without diabetes and 2282 participants with advanced CKD (eGFR < 30 mL/min/1.73 m^2^). At variance with previous studies, EMPA-KIDNEY enrolled 3192 patients (48.3%) with UACR <300 mg/g, 1328 of whom were normoalbuminuric. The study was stopped at a median follow-up of 2 years because empagliflozin demonstrated clear efficacy in reducing the risk of the primary composite outcome of kidney disease progression (decrease in eGFR of ≥40% from baseline, end-stage kidney disease or death from renal causes) and cardiovascular death (HR [95%CI] 0.72 [0.64–0.82]). As reported in the DAPA-CKD trial, the drug effect was similar in patients with and without diabetes. EMPA-KIDNEY also provided solid evidence that the effect of the drug was maintained in patients with severe CKD at baseline (eGFR of 20–30 mL/min/1.73 m^2^), finally confirming that even patients with advanced CKD benefit from SGLT2 inhibition. Compared to placebo, the drug reduced the outcome of kidney progression by 29% and significantly decreased hospitalization by any cause (HR [95%CI] 0.86 [0.78–0.95]). Previous studies suggested that, in diabetic patients, the nephroprotective effects of SGLT2is could be independent of the degree of proteinuria at the start of treatment [50]. In the EMPA-KIDNEY trial, however, some evidence of a larger risk reduction in patients with higher UACR was observed: the risk of reaching the primary endpoint was reduced by 23% with UACR > 300 mg/g, compared to a non-significant risk reduction of 9% in patients with UACR between 30–300 mg/g and a 1% increase in those with normal albuminuria. This effect, however, should be considered in the context of the relatively short trial period, which heavily reduced the number of events in patients at lower risk of progression. Indeed, when considering the long-term GFR reduction (i.e., the slope from 2 months to the last visit of the trial, after the initial dip in GFR caused by empagliflozin), the effect of SGLT2 inhibition was evident across all levels of UACR, with absolute differences of 1.76, 1.20 and 0.78 mL/min/1.73 m^2^ per year in favor of empagliflozin for each albuminuria class, respectively.

A recent collaborative meta-analysis of 13 RCTs led by the ‘SGLT2 inhibitor Meta-Analysis Cardio-Renal Trialists’ Consortium’ (SMART-C) was conducted to provide solid effect estimates of SGLT2i use in patients without diabetes (Table 1) [4]. The authors extracted summary-level data on 90,413 patients, both from available reports and unpublished information directly provided by trial investigators. Overall, treatment with SGLT2is reduced the risk of CKD progression (defined by sustained eGFR decrease ≥50% or < 15 mL/min/1.73 m^2^, end-stage kidney disease or death from kidney failure) by 37%, with a similar effect size in diabetic and non-diabetic patients. When the CDK trials were considered separately, the allocation to SGLT2is conferred protection from CKD progression of 40% in diabetic kidney disease, 30% in patients with ischemic/hypertensive kidney disease, 40% in patients with glomerulonephritis and 26% in patients with CKD of unknown etiology. Sensitivity analyses also suggested that this benefit was not modified by baseline renal function and albuminuria, and SGLT2is also conferred protection from acute kidney injury and cardiovascular events. Estimates of absolute benefit showed that the use of SGLT2is in diabetic patients led to a reduction in 11 kidney disease progressions, 4 episodes of acute kidney injury and 11 cardiovascular events per 1000 patient/years. The reduction in cardiovascular events was less marked in non-diabetic patients (−2 events per 1000 patient/years) as a consequence of the lower number of events accrued, but in this population the absolute benefit on renal events was maintained (−15 CKD progressions and −5 AKI episode per 1000 patient/years, respectively).

## 5. SGLT2is and Glomerulonephritis

SGLT2i trials with primary renal endpoints also included a large number of patients with glomerulonephritis as the cause of CKD, even though they excluded those who had received immunosuppressive therapy in the previous 3–6 months.

The DAPA-CKD trial enrolled 270 patients with IgA nephropathy (IgAN) on stable background RAS blockade. Only 14.1% of these patients were diabetic, and mean eGFR and UACR were 43.8 mL/min/1.73 m^2^ and 900 mg/g, respectively. In patients with biopsy-proven IgAN (94%), a prespecified secondary analysis found that dapagliflozin significantly reduced the risk of CKD progression, defined as a combined renal endpoint of eGFR decline > 50%, kidney failure or renal death (HR [95%CI] 0.23 [0.09–0.63]) [51]. As in the main trial, this effect did not differ across subgroups defined by baseline eGFR and UACR categories. Patients randomized to dapagliflozin experienced a reduction in UACR of 26% during the follow-up and a significantly slower CKD progression, with an eGFR difference between treatment arms of 2.4 mL/min/1.73 m^2^ per year after the initial dip resulting from SGLT2i initiation. The EMPA-KIDNEY trial protocol did not plan to conduct similar subgroup analyses in IgAN patients, but these were reported by SMART-C investigators in their meta-analysis [4]. With 817 patients diagnosed with IgAN enrolled in the study, EMPA-KIDNEY is the largest RCT conducted to date in this population. Combining results from EMPA-KIDNEY and DAPA-CKD showed a 51% reduction in the risk of CKD progression in IgAN. These results are likely to heavily impact not only clinical practice but also the design of future trials for interventions aimed specifically at IgAN.

DAPA-CKD and EMPA-KIDNEY also provided interesting, albeit less conclusive, information on the effect of SGLT2is in patients diagnosed with focal segmental glomerulosclerosis (FSGS). Among the 115 subjects included in the DAPA-CKD trial with FSGS reported as the cause of CKD, 104 (90%) had undergone bioptic confirmation and were included in secondary analyses [52]. At randomization, these patients had a mean eGFR of 42 mL/min/1.73 m^2^ and median UACR of 1248 mg/g, and there were comparatively more diabetic subjects in the placebo arm. Due to the relatively small number of events, the primary renal endpoint did not significantly differ between treatment arms. However, the between-group difference in UACR was 19.7% in favor of dapagliflozin, and the chronic eGFR decline rates were −1.9 and −4.0 mL/min/1.73 m^2^ per year in the dapagliflozin and placebo arm, respectively [52]. No data about proteinuria reduction or renal function decline are yet available from the EMPA-KIDNEY trial, but the primary renal outcome in the FSGS patients included in the study did not significantly differ between randomization arms (HR [95%CI] 1.24 [0.47, 3.25]) [4].

Notwithstanding the positive observations from DAPA-CKD subgroup analyses, these results should be interpreted with caution. Indeed, FSGS is a pattern of injury common to several pathogenic mechanisms that can be extremely different from each other in terms of short- and long-term outcomes. Not only did these trials also include patients without a bioptic confirmation of FSGS, which is arguably the only way to confidently make this diagnosis, but they were not designed to stratify patients according to the underlying FSGS form. Considering the pathogenic mechanisms, patients with secondary, maladaptive FSGS would be indeed more likely to benefit from interventions targeting renal hemodynamics such as SLGT2is. On the other hand, patients with primary, presumably immune-mediated forms or genetic FSGS may be less likely to respond to this approach. Since no data about baseline characteristics of these patients in the EMPA-KIDNEY trial are available, a direct comparison of FSGS patients included in the two studies is unfortunately not possible. It is tempting to speculate, however, that the UACR cap of 5000 mg/g included among criteria for enrollment in the DAPA-CKD trial might have reduced the proportion of patients with more progressive disease, i.e., those with severe nephrotic syndrome. Ad hoc trials with rigorous inclusion and stratification criteria as well as an adequately long follow-up will be required to assess the effectiveness of SGLT2is on kidney outcomes in these patients.

## 6. Adverse Effects of SGLT2is

Even though all the trials conducted on SGLT2is have clearly shown their favorable efficacy and safety profile, some potential adverse effects have been reported and should be considered when prescribing this class of drugs. 

By pooling the results of most of these trials, the SMART-C investigators have tried to address some of the major safety concerns [4]. Owing to their primary mechanism of action, i.e., the excretion of glucose in urine, it is unsurprising that one of the first safety signals reported was a higher rate in mycotic genital infections. Recent estimates confirm this notion and confirm that this adverse event is 3.57 times more likely in patients who are prescribed SGLT2is. Overall, there was also a significant, albeit modest (8%), increase in the risk of urinary tract infections, but the rate of serious events in this category was comparable between allocation arms. The CANVAS program originally raised an alert for the use of SGLT2is in diabetic patients due to a doubling of the risk of lower limb amputations. Although this was not confirmed by other trials considered separately and results from previous meta-analyses were inconclusive [53], pooled analyses conducted by SMART-C investigators revealed a 15% increase in the risk of this complication in diabetic patients. The risk was comparable between SGLT2is and placebo in non-diabetic individuals. Finally, the risk of euglycemic ketoacidosis in diabetic patients is more than doubled by SGLT2is, although the actual absolute number of events was very low (less than 0.3% of total patients developed one episode).

## 7. Future Perspectives

SGLT2is will likely continue to revolutionize the treatment of patients with chronic kidney disease in the foreseeable future. There is now compelling evidence of the beneficial effect of these drugs on cardiovascular and renal endpoints in CKD patients with diabetes or albuminuria, but several questions remain to be answered. Despite encouraging signals from RCTs, the benefit for non-diabetic patients without albuminuria still needs to be confirmed with more solid data, which would likely require studies of longer duration. Available evidence clearly supports the use of SGLT2is in CKD down to a GFR of 20 mL/min/1.73 m^2^, but there are currently no data for the efficacy and safety below this threshold. Several trials are under way to provide an answer to this question: the RENAL LIFECYCLES trial (NCT05374291) will assess renal and cardiovascular endpoints in 1500 patients with advanced CKD (eGFR < 25 mL/min/1.73 m^2^), transplanted (eGFR ≤ 45 mL/min/1.73 m^2^) or on maintenance dialysis (with significant residual diuresis). Several additional, smaller RCTs have been designed to focus specifically on the effects of SGLT2 inhibition in hemodialysis patients (NCT05179668 and NCT05614115), peritoneal dialysis patients (NCT05250752) or kidney transplant recipients (NCT04906213 and NCT04965935).

SGLT2 inhibition and RAS blockade will likely become the standard of care for patients with CKD, but their association with other therapeutic agents may provide additional benefits by targeting different components of CKD mechanisms. 

Several studies have shown that mineralocorticoid receptor antagonists (MRA) have an antiproteinuric effect in diabetic CKD patients (reviewed in [54]). More recently, the non-steroidal MRA finerenone has been shown to lower the risk of CKD progression and cardiovascular events in moderately proteinuric CKD patients with type 2 diabetes [55]. The therapeutic combination of SGLT2is with MRA on background RAS inhibitors has been explored in a pilot crossover trial that randomized proteinuric CKD patients to receive dapagliflozin, eplerenone or both drugs for 4-week intervals. The combined treatment resulted in an additive effect on proteinuria reduction, which was −19.6% for dapagliflozin alone, −33.7% for eplerenone alone and −53% for the combination [56]. The ongoing CONFIDENCE trial (NCT05254002) is expected to randomize 807 participants with CKD and proteinuria to finerenone, empagliflozin or both drugs, and assess the effect of the combined treatment on UACR. 

Finally, another class of drugs that has been attracting attention is that of endothelin receptor antagonists (ERA). These agents have been shown to reduce albuminuria in diabetic and non-diabetic patients with CKD, though at the cost of inducing sodium retention and edema [57,58]. The association of SGLT2is to ERA may result in a synergistic effect and reduce the adverse effect of sodium retention. To test this hypothesis, the ongoing phase 2 ZENITH-CKD trial (NCT04724837) will assess the effect of dapagliflozin and zibotentan on proteinuria in patients with CKD.

These studies are encouraging, as they could further extend the use of SGLT2is and identify patients in whom a combination of drugs with a multitarget effect may be more effective.

## Figures and Tables

**Figure 1 biomedicines-11-00279-f001:**
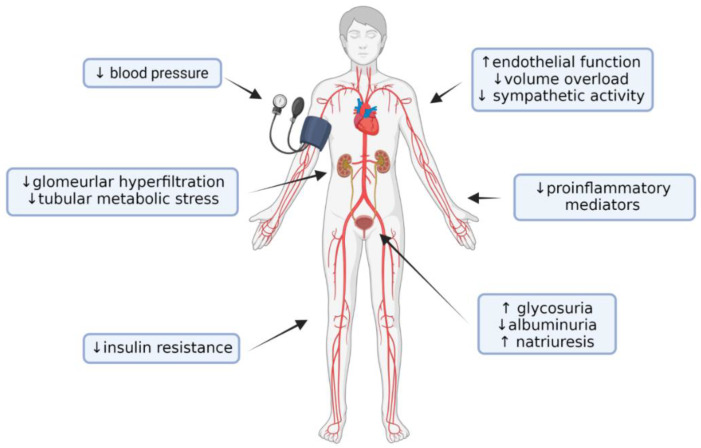
Beneficial systemic effects of SGLT2i with an impact on protection from CKD progression. Created with BioRender.com.

**Table 1 biomedicines-11-00279-t001:** Main outcomes of SGLT2i trials as summarized by SMART-C investigators [4]. Outcomes shown include kidney disease progression (sustained eGFR decrease ≥50% or < 15 mL/min/1.73 m^2^, end-stage kidney disease or death from kidney failure), acute kidney injury (MedDRA Preferred Term for AKI) and the combined endpoint of cardiovascular death and hospitalization for heart failure.

Randomized Controlled Trial	Kidney Disease Progression (RR [95%CI])	Acute Kidney Injury (RR [95%CI])	CV Death and HF Hospitalization (RR [95%CI])
**Diabetic Patients**			
EMPA-REG OUTCOME	0.51 [0.35–0.76]	0.41 [0.27–0.63]	0.66 [0.55–0.79]
CANVAS	0.61 [0.45–0.83]	0.66 [0.39–1.11]	0.78 [0.67–0.91]
DECLARE-TIMI 58	0.55 [0.39–0.76]	0.69 [0.55–0.87]	0.83 [0.73–0.95]
SCORED	0.71 [0.46–1.08]	1.04 [0.81–1.35]	0.77 [0.66–0.91]
VERTIS CV	0.76 [0.49–1.19]	0.95 [0.57–1.59]	0.88 [0.75–1.03]
DAPA-HF	0.73 [0.39–1.34]	0.79 [0.50–1.25]	0.75 [0.63–0.90]
EMPEROR-Reduced	0.52 [0.26–1.03]	0.77 [0.46–1.28]	0.72 [0.60–0.87]
EMPEROR-Preserved	0.82 [0.53–1.27]	0.69 [0.50–0.97]	0.79 [0.67–0.94]
DELIVER	0.87 [0.54–1.39]	1.13 [0.68–1.63]	0.80 [0.68–0.93]
SOLOIST-WHF	-	0.94 [0.55–1.59]	0.71 [0.56–0.89]
CREDENCE	0.64 [0.52–0.79]	0.85 [0.64–1.13]	0.69 [0.57–0.83]
DAPA-CKD	0.57 [0.45–0.73]	0.66 [0.46–0.96]	0.70 [0.53–0.92]
EMPA-KIDNEY	0.55 [0.44–0.71]	0.88 [0.64–1.20]	0.78 [0.60–1.03]
**Non-Diabetic Patients**			
DAPA-HF	0.67 [0.30–1.49]	0.60 [0.34–1.08]	0.73 [0.60–0.89]
EMPEROR-Reduced	0.50 [0.17–1.48]	0.56 [0.32–0.98]	0.78 [0.64–0.97]
EMPEROR-Preserved	0.68 [0.33–1.40]	0.80 [0.52–1.23]	0.78 [0.64–0.95]
DELIVER	1.01 [0.51–1.97]	0.64 [0.41–1.02]	0.82 [0.68–0.99]
DAPA-CKD	0.51 [0.34–0.75]	0.75 [0.39–1.43]	0.79 [0.40–1.55]
EMPA-KIDNEY	0.74 [0.59–0.95]	0.63 [0.41–0.97]	1.04 [0.65–1.67]

## Data Availability

Not applicable.

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
