# Peer review of "SGLT2 Inhibitors in Diabetic and Non-Diabetic Chronic Kidney Disease"

_biomedicines, 2023, doi:10.3390/biomedicines11020279_

Round 1

Reviewer 1 Report

The subject of the review is interesting since SGLT2i is a hot topic in nephrology. Although many reviews have been published on the issue, the quality of the submitted review is high since the authors provide the results of the main clinical trials with clarity and comprehensively. Also, data about the nephroprotective mechanisms of SGLT2i are provided. The manuscript is well-written. Generally, the review is written compactly and is quite informative.

Minor comment

Regarding the renoprotective mechanisms of SGLT2i, experimental studies also focus on the direct glucotoxicity on renal proximal tubular epithelial cells (Eleftheriadis et al., Int Urol Nephrol 2020; doi: 10.1007/s11255-020-02481-3 and Eleftheriadis et al., Int J Mol Sci 2022; doi: 10.3390/ijms232416107).

Author Response

Response to Reviewers (Reviewers’ Comments: bold; Authors’ Response: blue)

We would like to thank all the reviewers for taking the time to evaluate our manuscript and for their precious suggestions. Please find below a point-by-point response to the concerns raised.

Reviewer 1

The subject of the review is interesting since SGLT2i is a hot topic in nephrology. Although many reviews have been published on the issue, the quality of the submitted review is high since the authors provide the results of the main clinical trials with clarity and comprehensively. Also, data about the nephroprotective mechanisms of SGLT2i are provided. The manuscript is well-written. Generally, the review is written compactly and is quite informative.

Minor comment

Regarding the renoprotective mechanisms of SGLT2i, experimental studies also focus on the direct glucotoxicity on renal proximal tubular epithelial cells (Eleftheriadis et al., Int Urol Nephrol 2020; doi: 10.1007/s11255-020-02481-3 and Eleftheriadis et al., Int J Mol Sci 2022; doi: 10.3390/ijms232416107).

Thank you for highlighting this interesting point, which we failed to discuss in our original submission. We have updated the manuscript discussion to include evidence of SGLT2i nephroprotective effects linked to prevention of direct glucotoxicity on RTEC in the paragraph “Nephroprotective effects of SGLT2i”.

Reviewer 2 Report

In this manuscript, authors comprehensively reviewed the nephroprotective effects of sodium glucose transport protein (SGLT)-2 inhibitors in patients with non-diabetic chronic kidney disease, according to the recent randomized control trials. The topic of this review seems to be hot and interesting for many readers. The reviewer has some comments to the manuscript, as follows.

1. Regarding the mechanisms by which SGLT2 inhibitors exert nephroprotective effects, can ketone body production, which has been reported as one of the mechanisms of cardiac protection, be also involved in the beneficial effects? Incidence of euglycemic ketoacidosis actually increased with SGLT2 inhibitors treatment in patients with non-diabetic chronic kidney disease. Was it just adverse effect?

2. Authors summarized the results of several large randomized controlled trials. However, the reviewer recommends that authors should make a new table or figure showing the improved renal outcomes by each SGLT2 inhibitor therapy only in patients with non-diabetic chronic kidney disease.

3. In the section of adverse effects of SGLT2 inhibitors, many clinicians feel concern about the excessive initial dip of GFR and the development of acute kidney injury associated with SGLT2 inhibitor therapy in patients with chronic kidney disease. Authors should present the risk of acute decline in GFR with SGLT2 inhibitors in large clinical trials of chronic kidney disease patients with or without diabetes.

Author Response

Response to Reviewers (Reviewers’ Comments: bold; Authors’ Response: blue)

We would like to thank all the reviewers for taking the time to evaluate our manuscript and for their precious suggestions. Please find below a point-by-point response to the concerns raised.

Reviewer 2

In this manuscript, authors comprehensively reviewed the nephroprotective effects of sodium glucose transport protein (SGLT)-2 inhibitors in patients with non-diabetic chronic kidney disease, according to the recent randomized control trials. The topic of this review seems to be hot and interesting for many readers. The reviewer has some comments to the manuscript, as follows.

  1. Regarding the mechanisms by which SGLT2 inhibitors exert nephroprotective effects, can ketone body production, which has been reported as one of the mechanisms of cardiac protection, be also involved in the beneficial effects? Incidence of euglycemic ketoacidosis actually increased with SGLT2 inhibitors treatment in patients with non-diabetic chronic kidney disease. Was it just adverse effect?

We thank the reviewer for raising this issue, and agree with them that ketone body production may explain in part the cardioprotective effect of SGLT2i; evidence in the context of kidney benefit is unfortunately more limited in this regard, but it’s nonetheless extremely important to consider also this element when discussing the putative mechanisms involved in nephroprotection. We have added a brief discussion about this point in the  paragraph “Nephroprotective effects of SGLT2i”.

Regarding ketoacidosis, the SMART-C investigators have recently analyzed the incidence of this adverse event in patients without diabetes enrolled in SGLT2i-based RCTs, and found only one occurrence of this complication for approximately 30.000 participant-years of follow-up.

Altogether, we believe that ketone body production may have nephroprotective effects, but more studies will be needed to assess the relevance of this mechanism, especially in non-diabetic CKD patients.

  1. Authors summarized the results of several large randomized controlled trials. However, the reviewer recommends that authors should make a new table or figure showing the improved renal outcomes by each SGLT2 inhibitor therapy only in patients with non-diabetic chronic kidney disease.

According to the reviewer’s suggestion, we have created a table to summarize renal outcomes in SGLT2i trials; to harmonize the effect size of the trials, we decided to use the estimates provided by SMART-C investigators in their recent meta-analysis, which also allows to assess SGLT2i effects in diabetic and non-diabetic subjects separately. Of note, since another reviewer also made a similar suggestion, we included in the table the main details of all the studies cited in the manuscript, and included also a cardiovascular endpoint (CV death and HF hospitalizations).

  1. In the section of adverse effects of SGLT2 inhibitors, many clinicians feel concern about the excessive initial dip of GFR and the development of acute kidney injury associated with SGLT2 inhibitor therapy in patients with chronic kidney disease. Authors should present the risk of acute decline in GFR with SGLT2 inhibitors in large clinical trials of chronic kidney disease patients with or without diabetes.

We agree with the reviewer that many clinicians may feel anxious about the initial dip in renal function that follows SGLT2i initiation; this is not new in the nephrology field, as the use of RAS-inhibitors faced similar challenges in its prime time. In our opinion, data from RCT as well as from real-world experiences (10.1111/dom.13532) convincingly showed that the incidence of acute kidney injury is actually reduced in patients who are prescribed SGLT2i. Therefore, we strongly believe that these concerns should be promptly addressed to avoid stopping a medication that has demonstrated clear benefit. On the other hand, patient education is paramount to avoid volume depletion (e.g. “sick day” counseling, as nicely reviewed in 10.1016/j.ekir.2022.04.094) and potential complication of SGLT2i therapy. We have discussed this important point in the revised version of our manuscript.

Reviewer 3 Report

In this review article the authors' Podesta et al. have summarized results from randomized controlled clinical trials on a number of SGLT inhibitors in their efficacy against chronic kidney disease outcomes in diabetic and non-diabetic populations. The authors also summarize results from trials on improved cardiovascular with SGLT2i therapeutics. 

Overall, the authors have provided an excellent review on the current status of SGLT2i drug trials and their outcomes. They do address potential mechanisms of action and also list some limitation/precautions advised with the use of these therapeutics. 

Minor comments

1) It would be beneficial to tabulate the different trials listed with the most important outcome within the context of this review. 

2) Textual edits in lines 119, 129, 135, and 178

Author Response

Response to Reviewers (Reviewers’ Comments: bold; Authors’ Response: blue)

We would like to thank all the reviewers for taking the time to evaluate our manuscript and for their precious suggestions. Please find below a point-by-point response to the concerns raised.

Reviewer 3

In this review article the authors' Podesta et al. have summarized results from randomized controlled clinical trials on a number of SGLT inhibitors in their efficacy against chronic kidney disease outcomes in diabetic and non-diabetic populations. The authors also summarize results from trials on improved cardiovascular with SGLT2i therapeutics. 

Overall, the authors have provided an excellent review on the current status of SGLT2i drug trials and their outcomes. They do address potential mechanisms of action and also list some limitation/precautions advised with the use of these therapeutics. 

Minor comments

1) It would be beneficial to tabulate the different trials listed with the most important outcome within the context of this review. 

We thank the reviewer for this suggestion; we have included a table summarizing the most important outcomes of the RCT listed in the revised version of our manuscript (please also refer to the response to Reviewer 2).

2) Textual edits in lines 119, 129, 135, and 178

Thank you for catching these typos, we have corrected them.

Round 2

Reviewer 2 Report

Authors have successfully addressed all of concerns in the revised manuscript. There are no more comments.